# A Self-Contained 3D Biomechanical Analysis Lab for Complete Automatic Spine and Full Skeleton Assessment of Posture, Gait and Run

**DOI:** 10.3390/s21113930

**Published:** 2021-06-07

**Authors:** Moreno D’Amico, Edyta Kinel, Gabriele D’Amico, Piero Roncoletta

**Affiliations:** 1SMART Lab (Skeleton Movement Analysis and Advanced Rehabilitation Technologies)—Bioengineering & Biomedicine Company, 65126 Pescara, Italy; damico.mandelli@vodafone.it (G.D.); bbcmail@tin.it (P.R.); 2Department of Neuroscience, Imaging and Clinical Sciences University G. D’Annunzio, 66100 Chieti, Italy; 3Department of Rehabilitation, University of Medical Sciences, 61-545 Poznan, Poland; ekinel@ump.edu.pl

**Keywords:** stereo-photogrammetry, baropodometry, posture, 3D spine, skeleton model, gait analysis, running analysis, movement analysis, surface electromyography

## Abstract

Quantitative functional assessment of Posture and Motion Analysis of the entire skeleton and spine is highly desirable. Nonetheless, in most studies focused on posture and movement biomechanics, the spine is only grossly depicted because of its required level of complexity. Approaches integrating pressure measurement devices with stereophotogrammetric systems have been presented in the literature, but spine biomechanics studies have rarely been linked to baropodometry. A new multi-sensor system called GOALS-E.G.G. (Global Opto-electronic Approach for Locomotion and Spine-Expert Gait Guru), integrating a fully genlock-synched baropodometric treadmill with a stereophotogrammetric device, is introduced to overcome the above-described limitations. The GOALS-EGG extends the features of a complete 3D parametric biomechanical skeleton model, developed in an original way for static 3D posture analysis, to kinematic and kinetic analysis of movement, gait and run. By integrating baropodometric data, the model allows the estimation of lower limb net-joint forces, torques and muscle power. Net forces and torques are also assessed at intervertebral levels. All the elaborations are completely automatised up to the mean behaviour extraction for both posture and cyclic-repetitive tasks, allowing the clinician/researcher to perform, per each patient, multiple postural/movement tests and compare them in a unified statistically reliable framework.

## 1. Introduction

Across biomechanical and clinical research literature, the interest in spinal and postural associated pathologies and the assessment of their induced functional disability are widely represented. Functional analysis quantification is essential for a valid assessment of impairment, the design of treatment plans [1], the surgery planning [2], the monitoring of the progression of pathology and/or treatment outcomes [3,4]. Part of the scientific literature that deals with spinal disorders and deformities, particularly the surgical–orthopaedic one, historically focused only on what happens at the trunk, primarily considering the coronal plane only [5,6,7,8]. More recently, there has been an increasing interest in the sagittal plane spine and pelvis alignment, especially when dealing with adult spinal deformity (ASD) [9,10,11]. Advances in ASD assessment through novel full-body imaging technology [12] accentuate the need for head-to-toe radiographic evaluation and the visualization of the lower extremity compensatory mechanism, including knee and ankle flexion, hip extension and pelvic displacement [13]. The advancement in image-processing-based diagnostic technology (e.g., digital X-ray, digital 3D stereo reconstruction of X-ray [12], CAT scans, MRI and lately, digital 3D ultrasound techniques [14,15,16]) brought a significant increase in the assessment of skeletal anatomical structures and spine-related pathologies. Nevertheless, except for dynamic X-ray and the very recent dynamic MRI, none of these methods can provide details on the vertebral column’s functional condition [17,18,19]. Other approaches, such as the rasterstereographic back-surface measurement technique or the recently introduced ultrasound measurement, raise concerns and questions which need further clarification [16,20,21,22,23] about measurement accuracy and/or the need for the patient to keep a constrained position during the scanning measurement.

Considering the above limitations, a technique called optoelectronic stereophotogrammetry, increasingly documented in the literature, has been shown to provide an unobtrusive and harmless yet complete significant solution for capturing full-skeletal biomechanical characteristics, including the 3D shape of the spine, providing the functional appraisal needed to address clinical problems in rehabilitation medicine [1,17,18,19,24,25,26,27,28,29,30,31,32,33,34,35,36,37,38,39,40,41,42,43,44,45,46,47]. 

Among such original 3D optoelectronic stereophotogrammetric approaches, only a few presented a fully detailed 3D modelling of the spine. Indeed, various solutions have been developed for the modelling of the vertebral column. The first line of research focused on representing the spine in various rigid segments (including multiple vertebrae) identified by clusters of markers [39,40,41] (see Needham et al., 2016 for a review [47]). Another approach was to model the column by linear segments joining the markers placed on spinous processes (usually one out of two) [31,32,33]. Finally, one last approach has been proposed in which the positions of the markers on the spinous processes (mainly one every second spinous process along the spine with few exceptions) have been used as a basis for a polynomial interpolation [34,35,36,37,38], or fitting circumferences [42,43,44] to reconstruct the spine shape and assess its curvature. However, a significant limitation of such polynomial-based methods concerns the 2D nature of the information extracted. Indeed, despite applying the same interpolation for both the sagittal and frontal planes, the method provides a separate curve for each plane (and not a 3D curve) and no information concerning spinal rotations [36].

Conversely, D’Amico et al. used spline interpolation of 11 spinous processes’ 3D positions (from C7 down to S3 every second vertebra), improving an original method presented in Bryant et al. [30]. The improvement was reached by smoothing the noisy interpolated spine measurement using sophisticated automatic adaptive digital filtering [17,19,24,25,26,48,49] to obtain the full spine shape’s mathematical model as one single curve in the 3D space. Therefore, unlike the methods based on polynomials, with this latter mathematical approach, the frontal and sagittal profiles of the spine are obtained by suitably projecting the shape of the 3D spine, respectively, on the frontal and sagittal planes of the subject. The influence of the foot attitude on the spine and pelvis postures has been investigated in the literature. Notably, low back pain [28,50,51,52] and the role of foot orthoses in the prevention and care of spine disorders have been studied. Results appear to be controversial, probably due to differences in the measurement approaches used, some of the methods being insufficient to capture and describe the complexity of entire-body posture.

This paper proposes an instrumental and methodological approach to integrating baropodometric information with the entire skeleton’s kinematic data, including the 3D reconstruction of the spine, to evaluate either static postures or whole-skeleton movement comprehensively during gait or various functional tasks. In this form, the developed methodology represents a novelty in the literature.

Indeed, except for a few examples, the study of spine biomechanics and baropodometry has had independent paths even if their close connection is evident.

Plantar pressure analysis has been used in clinical practice since the early 1980s, when reliable pressure measurement devices (PMDs) became available [53,54]. However, over the years, PMDs have been primarily used as standalone measurement devices. Since the early 2000s [55], the integration of PMDs with stereo-photogrammetric systems and ground reaction force plates have been presented in the literature. Moreover, the primary focus has mainly devoted to the study of the particularised biomechanics of the foot [56,57,58,59,60,61,62,63] more than the full-body biomechanics (see Giacomozzi et al., 2018 for a review [53]). Not one of such studies considered the use of a baropodometric treadmill for pressure data collection, and except for a PMD prototype [55], technical constraints about synchronisation between PMDs and kinematic systems have been presented in all such previous works. As a matter of fact, the kinematic and pressure data have been limited to ‘simultaneous’ and not fully synched, triggered acquisition. Some offline software-based subsequent spatial and/or temporal data realignment through signal processing has been necessary.

To overcome such limitations, a new multi-sensor system named GOALS-E.G.G. (Global Optoelectronic Approach for Locomotion and Spine Expert Gait Guru) (Bioengineering & Biomedicine Company Srl—Pescara, Italy), based on Optitrack hardware (NaturalPoint Inc.—Corvallis, OR, USA) integrating an optoelectronic stereophotogrammetric system, a baropodometric treadmill and a telemetric surface electromyographic device (SEMG) has been developed.

The aim was to achieve a multifactorial analysis of posture and movement, integrating synched 3D full skeletal kinematics (including spine), baropodometric assessments and electromyographic data. 

This was achieved by merging validated models of the lower limbs described in the literature into the 3D parametric biomechanical skeleton model (upper limb not yet included) developed by D’Amico et al. [17,18,19,24,25,26,27,28,29] and by extending the features of the latter to kinematic and kinetic analyses of gait and run. 

In this way, it is possible to evaluate postural anomalies of upright posture and study how they can induce alterations in dynamic motor tasks such as walking, flexion-extension, lateral bending, rotations and/or other movements of the trunk and cervical spine. 

Full validation for the 3D spine mathematical modelling in dynamic conditions is provided. Furthermore, a paradigmatic example of the quantitative functional description of a pathological case will be discussed at length in the Results section to show this multifactorial approach’s capability to process many different measurements into a unified view, merging information from both the static posture and gait analyses. 

## 2. Materials and Methods

This project stems from over 30 years of biomechanical-clinical research following five specific goals:(1)Describe the 3D whole-body posture with a comprehensive reconstruction of the 3D spine shape to investigate spine disorders or how the spine is involved in the movement or locomotion in healthy subjects and orthopaedic and neurological pathologies.(2)Provide full integration and data fusion between kinematic and baropodometric data.(3)Obtain the largest possible number of clinically relevant parameters in functional motor tasks evaluations employing the stereo-photogrammetric approach, while using the smallest number of body landmarks to keep the evaluation procedure the most straightforward possible for both patient and clinical operator.(4)Provide the data elaboration up to clinical results representation as automatic as possible and algorithm-based, leaving the operator the task to check data integrity.(5)Allow biomechanical analysis of gait and movement in delimited areas. Indeed, the clinical structures often present logistical problems in finding a suitable and ample space to build a laboratory for analysing posture and gait. For this reason, it was decided to adopt a treadmill as a basis for gait assessment.

To meet the above criteria, we expanded to dynamic movement the adjustable mathematical–biomechanical model of the body skeleton previously developed for posture analysis [17,18,19,24,25,26,27,28,29].

Several innovative methodological approaches are introduced and explained: a new 3D spine mathematical modelling for static and dynamic analysis, that allows together with a trunk model and baropodometric measurements to assess spine loads at an intervertebral level; a new averaging process for any kinematic variable of interest performed through a specially developed normalisation and smoothing procedure; a new averaging process for baropodometric data; a new statistical approach to perform side-to-side and pre–post treatment point-by-point comparisons for each kinematic variable; a new statistical score named the index of estimated differences (IED) to summarise in percentage terms the statistically different proportions in the comparison between the average temporal courses of pairs of congruent variables.

### 2.1. Hardware Configurations

Depending on the specific analysis to perform, the GOALS-EGG system can be configured in different settings. In any case, a minimum of 12 IR (infrared) TVCs (TV cameras) with a 0.3 Mpix resolution are required to perform full-skeleton gait and run analyses safely. The chosen family of treadmills is the Zebris-FDM-T^3^, presenting embedded baropodometric Zebris FDM (ZEBRIS Gmbh, Isny, Germany) platforms in a wide selection of sizes, sampling frequencies, sensor numbers and resolutions. The manufacturer grants the accuracy of ±5% of the maximum range on the calibrated pressure measuring range (1–120 N/cm^2^).

For the case study chosen as a paradigmatic example (presented in the Results section), the adopted system configuration was: a 0.3 Mpix 12 IR TVCs GOALS kinematic system and a Zebris FDM-TS70L-3i baropodometric treadmill equipped with an embedded platform measuring 94.8 cm × 40.6 cm, 5376 sensors, 1.4 sensors/cm^2^ (Figure 1). Both devices are driven and synchronised by a software-controlled external genlock trigger (through a specially developed software-controlled variable frequency PWM signal generator) at a 100 Hz maximum sampling rate. The configuration is sufficient to allow the automatic analysis of the upright standing posture, gait, running and other neck and trunk functional motor tasks. 

The standard volume of acquisition, i.e., the actual calibrated volume within which the subject can be measured with established precision and consistency, is about 3 m long by 3 m deep by 2.5 m high. The standard final mean 3D stereo-photogrammetric error, in such a working volume, presents an upper limit of 0.3 mm [24,25,26,27].

An essential step of the calibration procedure is related to establishing the relative position of the baropodometric platform (embedded in the treadmill) into the calibrated volume. This position is essential to proceed to all the following calculations related to underfoot pressure maps and associated vertical forces, joint forces and torques. The baropodometric platform constitutes the support surface of the treadmill belt. The ground plane is established on the treadmill walking surface during the stereo-photogrammetric calibration step once such a surface has been adjusted to be perfectly horizontal. The calibration procedure defines the laboratory reference system. Its origin is placed on the top-left platform vertex. The baropodometric platform is rectangular, and the X and Y axes are defined along platform edges, the Y-axis along the shorter edge and the X-axis along the longer edge. Following the right-hand rule, the Z-axis is orthogonal to the ground plane (i.e., in the direction of the gravity line). A telemetric system (Wave Plus wireless EMG—Cometa srl, Bareggio (MI) Italy) records surface electromyographic (SEMG) signals following the guidelines of the SENIAM European project [64] for electrodes positioning. The lower limb muscles activity is recorded to investigate motor coordination/dysfunction in gait analysis. Conversely, bilateral activities in multifidus and erector spinae–longissimus dorsi activities allow the investigation of the flexion relaxation phenomenon in low back pain [65,66]. In patients with neck pain, SEMG on the upper trapezius and sternocleidomastoid muscles is usually recorded bilaterally. 

### 2.2. Scalable Biomechanical Skeleton Model and Acquisition Protocols

After developing crucial signal processing procedures for smoothing and derivative assessment [39,40], in the mid-1990s, our group started developing a complete and precise 3D biomechanical model of the human skeleton (upper limbs not yet included) by combining different segmental biomechanical models presented in the literature [17,18]. The focus was to study the entire human skeleton posture and function, and scrupulous care has been devoted to the exact reconstruction of spinal detail. 

The model’s accuracy and precision are founded on in-house original signal processing and optimisation procedures [17,18,19,28,29,48,49,67,68] and anatomical studies listed in the literature (cadaver dissections, in vivo X-ray and parametric regression equations from gamma-ray measurements) [69,70,71,72]. Furthermore, the model was formulated in a parametric form to allow for scaling any subject’s characteristics by fitting each given skeletal segment to the 3D measured positions of its corresponding body landmarks [17]. 

Various protocols involving different body labels allow scaling of the biomechanical model complexity. The analysis’ purposes and requirements determined the best choice among the available protocols [17,19,24,25,26,27]. 

To allow in a single acquisition session the complete analysis of upright standing posture, spine functional evaluation and gait or running biomechanical characterisation, we herein propose an extension of the 27 markers ASAP 3D protocol [73] originally developed for 3D posture and spine-related pathologies (scoliosis, back pain, etc.) evaluation. Two different protocols have been developed. Both share the same set of landmarks in the head, trunk and pelvis used for posture evaluation. In the first protocol, employing a 31-markers set, the legs’ labelling is derived by the very well-known protocols of Davis et al. and Kadaba et al. [74,75], but where the ASIS and PSIS (anterior and posterior iliac spines) positions provide the basis for the assessment of hip joint centre positions and of pelvis width [69]. The comprehensive list of these anatomical landmarks and an example of the entire 3D skeleton reconstruction (upper limbs not yet included) obtained via opto-electronic stereo-photogrammetric measurement is given in Figure 2. 

However, it has been shown that Davis et al. and Kadaba et al. [74,75] models can suffer from some limitations in out-of-sagittal plane rotations, especially at the knee and ankle joints, although good correlations were observed for most of the gait variables when compared to others models [76]. For such a reason, when a more in-depth focus on out-of-sagittal plane rotations is of interest, a second more complex protocol based on the Rizzoli marker set [77] increases the used bony landmarks up to a total of 49 to better describe the pelvis–lower-limb apparatus modelled as a 9-link chain (pelvis, thigh, shank and foot) [19]. Differently from the description given in Leardini et al., 2007, the medial femoral epicondyles and medial malleoli positions are directly measured, during the anatomical calibration procedure, by placing markers on such landmarks instead of using a pointer [77]. In both the lower-limb models, some additional technical markers (out of anatomical landmarks) are added to simplify body tracking and automatic marker labelling as well as optimisation procedures [78]. Compared to the Rizzoli marker set, an additional marker is placed on the tip of each hallux to model the foot with two segments: the rear-foot identified by the markers on the heel and first and fifth metatarsal heads; the forefoot identified by the markers on the first and fifth metatarsal heads and the tip of the hallux. A cylindrical joint connects the two-foot segments with its axis through the first and fifth metatarsal head. A further detailed description of such a model is beyond the limit of the present paper.

Due to skin movements, the markers placed on lower limbs generally displace, rotate, and eventually deform relatively to the underlying bone. This phenomenon is known as soft-tissue artefacts. In the skeleton model, the global optimisation method proposed in the literature [78] to reduce such phenomenon has been implemented. The optimisation considers all segments and joints of the 9-links chain model simultaneously, minimising the overall differences between the measured and model-determined marker coordinates in the least-square sense. 

When the investigation focuses on neck and back pain, different test batteries have been developed to assess the associated postural and spinal dysfunction [25,65]. To this aim, the marker sets are modified using three additional markers placed on a headband (Figure 2). Thus, the head and neck are reconstructed even during a forward-bending test [25,65] when the face’s markers disappear to the cameras’ view. Head rotations can also be assessed in the same manner for patients with neck pain. The head anatomical reference system is defined by the three markers placed on the zygomatic bones and the chin. Such markers define the YZ head frontal plane. In this plane, the Y mediolateral axis is given by the line joining the zygomatic bone pointing from right to left. The Z is orthogonal to Y-axis, pointing up. The X-axis is determined by the right-hand rule pointing forward. The origin of such a reference system is placed on the mid-point between the zygomatic markers. 

The instrumented treadmill allows the recording of underfoot pressure maps and the vertical component of ground reaction forces during the foot–floor interaction. As the baropodometric platform’s position is calibrated in the laboratory reference system, it is possible to calculate the centre of pressure (COP) position (i.e., the application point of the vertical ground reaction forces) per foot. In this way, the lower limbs joint net forces and torques estimates are solved using Newton’s motion equations in an inverse dynamics problem. The distribution of loads between the lower limbs and each foot’s support modalities are evaluated. The presence or absence of a flat foot, hollow foot, pronation or supination, anterior or posterior imbalances and patterns of the COP (oscillations, spatial-temporal evolutions) for each foot are appraised and correlated with the overall attitude in the standing position or during walking/running. The joint net forces, torques and muscle power at each lower-limb joint can be assessed by considering the ground reaction forces and the body’s segmental inertial properties [69,70,71,72,79,80,81]. The baropodometric embedded measurements can be used for the automatic gait/run-time-events identification given the hardware genlock synchronisation between the treadmill and the stereophotogrammetric system. In this way, even stance phases sub-events (i.e., heel contact and load acceptance, forefoot contact, rockers and push-off phases) can be automatically identified by segmenting the foot pressure maps into foot sub-regions (i.e., rearfoot, midfoot, forefoot). It is worth noting that the stance phase sub-events automatic identification algorithm works appropriately when the subject is able to place the foot entirely on the ground during the support phase. Specific algorithms based on template recognition are under development to handle patients with foot deformities (i.e., equine foot, crouch knee). 

The elaboration has also been extended to assess net forces and torques at each spine intervertebral level by integrating, in the skeleton biomechanical reconstruction, the trunk model by Liu and Wickstrom 1973 [72]. In the latter, the trunk is subdivided into 17 per-vertebrae transversal slices starting from T1 down to L5. For each slice, parametric regression equations for estimating the distribution of the human torso’s inertial properties are derived from segmented cadaveric data. In particular, the slice mass, the slice centre of mass position relative to the spinous process position, and the moment of inertia are provided. The most distal segment of the T1 trunk slice is defined by considering the cervical vertebrae and skull as a single rigid body for simplification. The inertial parameter for such distal segment is derived from Zatsiorksy 1990 [71]. The L5 vertebra slice is the most proximal one and interacts with the pelvis segment. As the upper limbs are not yet included in the skeleton model, their masses are considered to be concentrated entirely around the marker positioned on the corresponding shoulder’s acromion and are considered in assessing the net forces and torques acting at each intervertebral disc. These load stresses are derived by solving an inverse dynamic problem where the computation starts from the most distal segment [81,82]. 

### 2.3. Biomechanical Data Acquisition and Measurement Processing

#### 2.3.1. 3D Posture Analysis

The typical trial session starts with the evaluation of the subject’s static standing posture. The standing posture is evaluated in different conditions [24]: the indifferent orthostasis (IO) (i.e., the neutral erect standing); the self-correction orthostasis (SCO) (i.e., when the subject is asked to assume the best correct self-perceived standing posture); and when necessary, the wedge-corrected orthostasis (WCO) (i.e., when the subjects present a leg-length discrepancy (LLD) that needs to be corrected) [24]. Each static postural attitude is correctly identified when at least 5 2-s acquisitions are performed. For the chosen system configuration, a minimum of 1000 measurements are averaged per each static postural stance. 

Different positions of the feet can influence standing posture. Thus, the subject is asked to align heels on a line parallel to the Y-axis of the laboratory reference system and keep feet apart (without restricting feet directions) at about pelvis width (i.e., with feet under the hip joints’ projection). Real-time baropodometric measurement availability allows controlling feet alignment straightforwardly by checking that the heels’ most prominent tip, in the foot pressure maps, lay on the same row. In case of feet deformities and/or when heel contact is not possible (i.e., equines foot, crouch knee) the feet position can be aligned considering the toes.

Before averaging, an amount of pre-processing is needed on the acquired 3D raw data to define the subject’s local coordinate system and its orientation relative to the global coordinate system [17,19,24,25,26,27]. We used the general definitions provided by the Scoliosis Research Society [83]. However, in distinction to such recommendations, PSIS rather than ASIS landmarks (see Figure 2) have been considered in defining the subject’s local coordinate system to reduce propagation errors and other interference deriving from pelvis torsion in the subsequent calculation of spinal parameters. Therefore, in a right-handed system: the frontal-coronal (YZ) plane is the vertical plane containing the PSIS. In such a plane, the Z-axis is the vertical axis pointing up from the mid-point between the PSIS. The Y-axis is orthogonal to the Z-axis, passing through the mid-point between the PSIS, pointing from the body’s right side to the left. The X-axis is determined by the right-hand rule passing through the mid-point between the PSIS pointing forward, i.e., from the back to the body’s front. The XZ axes define the sagittal plane; the XY axes define the horizontal plane [24]. Once the subject’s local coordinate system has been defined, its origin is translated into the S3 position for anatomical convenience. Once having determined this individual system, a rotation is performed within each frame to align the subject’s coordinates with the global reference coordinates. Once the alignment is complete, it is possible to average all acquired frames properly. Based on measurements of the 11 3D spinous processes, data are interpolated using cubic splines [84] in order to assess the position of each unlabelled spinous process and intervertebral disc. After interpolation, the space-curve modelling of the spine is analytically represented using three parametric functions x(t), y(t), z(t) (the parameter being t > 0). A smoothing and differentiation procedure specially developed for interpolated data with cubic splines is applied to these functions [17,48,49,67]. Once the three parametric functions x(t), y(t), z(t) are identified, it is possible to define a spine model estimating the 3D position of each vertebra from C7 down to S3. To define a local anatomical reference system for each vertebra, we followed the indications of Wu et al. [85]. The intervertebral articulations have six degrees of freedom (three translations and three rotations), each of which has a measurable stiffness. Therefore, there are six independent parameters of motion (three displacements and three rotations). In our model, the local vertebra anatomical reference frame is defined by the moving trihedron [86] identified by the tangent, normal and binormal of the analytical 3D space-curve in each point. Since this space-curve is identified by the markers glued to the spinous processes, the origin of the so-defined local vertebra anatomical reference frame is placed on the vertebral spinous process instead of the centre of the vertebral body as described in Wu et al. [85]. The tangent, binormal and principal normal vectors of the curve are calculated at each point by applying the Frenet–Serret formulas [86] starting from the first and second derivative of the parametric equations of the 3D curve assessed through the above-mentioned specially developed smoothing and differentiation procedure [17,48,49,67]. 

When the reconstruction of the 3D skeleton is obtained, all the clinical parameters usually measured on the radiographic image are automatically determined to quantify spinal morphology (i.e., Cobb and Kypho–Lordotic angles either in the frontal or the sagittal planes, respectively). The signal’s derivatives (identifying the tangents to the curve in each point) are computed from the specially developed digital filtering procedure. The maxima and minima of the assessed first derivative allow selecting, under analytical constraint, all and only the inflexion points defining the limit vertebrae. After determining the limit vertebrae, the Cobb and Kypho–Lordotic angles computation is straightforward, computing the angle between the tangents in such points. Besides, a collection of relevant biomechanical variables are extracted [17,19,24,25,26,27]. For example, spinal offset and global offset describe trunk and global unbalancing in either the frontal or the sagittal planes. The spinal offset is defined as the displacements of each spine marker relative to the vertical line passing through the S3 vertebra, and the global offset is defined as the displacements relative to the vertical line passing through the middle point between the markers located on the heels. Both the spinal and global offsets are computed in the above-defined subject’s local frontal and sagittal planes. Finally, the averaged global and averaged spinal offset values are computed by averaging the offsets of each vertebra to obtain summarising descriptive parameters. Other parameters include pelvis frontal and sagittal inclinations, pelvis torsion, shoulder-to-pelvis, pelvis-to-heels and shoulder-to-heels horizontal rotations, underfoot load distributions. An example of the described parameters can be found in the next results section.

#### 2.3.2. 3D Trunk and Neck Functional Tests

For dynamic analyses (functional trunk and neck flexion–extensions, lateral bending and axial rotations tests, etc.), when significant trunk torsions are present, the subject’s local coordinate system can be complemented by an additional anatomical local coordinate system for the thorax. For example, in order to minimise possible projection error due to the transverse plane rotation of the trunk during axial rotations or lateral bending (or, in some case, also during gait/running), the thoracic curves can be calculated using a dynamic coordinate system of the thorax based on ISB recommendations [87]. Indeed, for example, if the subject performs a thorax axial rotation of 90 degrees with the pelvis kept fixed, the natural sagittal kyphotic curve will appear as a deformity in the frontal plane of the pelvis-linked reference frame. The above-described error can be avoided by defining an additional thoracic-linked reference frame that follows the axial thorax rotations. In this way, a thorax-moving sagittal plane is defined using the markers placed on the sternum and two spinous processes, i.e., the plane formed by C7, the middle point between the left and right sternoclavicular joints and the midpoint between Xyphoid and T8. Moreover, such a thorax reference system is always used in relation to the skull’s reference system (based on the markers placed on zygomatic bones and chin, Figure 2) when the neck mobility is analysed through functional tests. 

#### 2.3.3. 3D Analysis of Walking and Running

The gait and run are a cyclic sequence of stances, phases and flight phases during the body’s forward transfer. Despite their cyclical nature, how the support phases follow one another always presents minimal differences in terms of temporal duration, kinematic of movement and foot–floor interaction loads. Such intrinsic variability connected to gait/run, both healthy and pathological, implies the necessity to approach the study of its characteristics from a statistical point of view, namely, by defining a mean gait/run cycle (mean stride) and associated variability to extract clinically relevant parameters with a statistical significance. Before the gait or running session starts, participants are afforded a treadmill acclimatisation period, in which they are briefed regarding the safety procedures and provided with a 10 min familiarisation time of walking/running at a self–selected speed. The data collection for gait or running is in general obtained with a single acquisition long enough to ensure that at least 5 to 10 full strides are recorded, i.e., a number sufficient to guarantee the extraction of an average cycle that can be considered representative from a statistical point of view. An important goal was that the average cycle could be computed automatically. The determination of the mean gait cycle is a complex process that is divided into several steps. At first, it is necessary to determine the stride events. Given the hardware synchronisation between the baropodometric treadmill and the kinematic system, the gait events identification is accomplished automatically by classifying foot contacts (i.e., heel strike and toe-off) through baropodometric data.

The mean stride sought is the composition of the sequence of support and flight phases conveyed on a common time base. As specified above, the intrinsic gait variability implies that each stride lasts differently from the others. Given the measurement-fixed sampling rate, the number of samples will be different from stride to stride. The common time base is obtained by computing the average stride time duration. Then, for each determined stride, each variable’s time courses are resampled to the average number of samples through a specially adapted version of the interpolation-smoothing procedure presented in the past [48,49,67]. The signal processing mathematical details are beyond this article’s scope. This collection of procedures is defined as time normalisation. Once the time normalisation is completed, the averaging can be appropriately performed for each variable of interest resampled time course, and mean stride is derived. In this way, the averaged time course and associated standard deviation are defined per each variable of interest. 

It should be noted that the normalisation and averaging processes in walking and running differ only because the double support phases are missing in the running. 

Indeed, once the mean gait/run cycle is obtained, point-to-point paired *t*-Tests comparing congruent variable’s time courses (e.g., time courses of right vs. left hip joint angle, etc.) are applicable. In this way, it is possible to statistically describe either the different side-to-side behaviours (i.e., differences between the left and right legs) or evaluate changes in the gait biomechanical patterns measured in different conditions (e.g., induced by some intervention like pre–post therapy). Thus, it is possible to identify all the sub-regions in which the variables of interest’s temporal course present differences.

A preliminary ‘similarity speed’ check and walking or running steady-state conditions check is performed in pre–post therapy comparison. Comparison is not allowed if the two walking/running speeds differ by more than 5%, and the steady-state conditions are not verified.

A modified version of a numerical index named the ‘Index of Estimated Differences’ (IED), introduced by Santambrogio in 1989 [88], was developed to summarise in percentage terms the statistically different proportion in the comparison between the average temporal courses of pairs of congruent variables, in the normalised mean gait/run cycle. Thus, an IED value = 100% in the left vs. right leg comparison of the mean course of the hip joint angle means that they are statistically different at each point. 

### 2.4. Baropodometric Analysis of Walking and Running

Particular care has been devoted to developing algorithms for baropodometric data elaboration and their subsequent use with kinematic–kinetic variables into the biomechanical skeleton model. The first step was to appropriately determine the baropodometric measure of plantar support during walking on the treadmill. In fact, considering the sliding of the treadmill belt (and therefore of the feet) relatively to the underlying baropodometric platform, it was necessary to create an algorithm that would match the sensors of the platform with the anatomical points of the sole of the foot for each frame. This was achieved by implementing a special signal processing and smoothing algorithm to determine the belt’s instantaneous speed. In this way, it was possible to establish the relative sliding of each foot on the platform by which it was possible to reconstruct the pressure contribution for each underfoot area. Thus, by mathematically removing the relative movement between the feet and the platform, it was possible to construct a virtual grid of pressure sensors fixed to the sole of the foot in which each cell of the grid corresponds, for each frame, to the same anatomical position under the foot. Subsequently, the elaboration procedure applies to baropodometric data the same approach for mean gait-cycle computation defined for kinematic variables. In this case, the time resampling–normalisation–averaging procedures have been applied on the time courses of pressures per each sensor of the above-defined foot-fixed virtual grid. Once the so-defined baropodometric mean stride has been determined, it is possible to extract all the averaged spatio-temporal gait parameters. Besides, the mean vertical forces and the mean time-varying underfoot pressure maps are computed for each foot during gait.

Furthermore, a new computational feature has been introduced to evaluate the underfoot pressure-distribution symmetry. Such symmetry is evaluated considering the mean peak pressure maps (i.e., the grids built up with the highest plantar pressure values determined during the mean stance phases in every single cell) and calculating the difference between anatomically corresponding cells’ pressure values of the two feet. An example of such difference maps computed for the presented case study is given in the results.

### 2.5. Error Analysis of 3D Spine Reconstruction Algorithm through Numerical Simulation

Spine shape variations are obtained per each recorded frame using a specially developed improved version for dynamic tests of the signal processing procedure validated for posture analysis [24]. Several numerical tests have been performed to validate such an improved version. Changes of the spine shape during dynamic tests have been simulated by a set of 11 noisy synthetic helixes at different amplitude and period (considering the trunk size of an adult 1.75 m tall), in the following form: x(i) = A[j]*sin(*π**f[j]*i/41)     1 ≤ i ≤ 41y(i) = A[j]*cos(π*f[j]*i/41)       1 ≤ i ≤ 41z(i) = k*i              1 ≤ i ≤ 41(1)

Each analytical helix is evaluated in 41 points (i.e., the total number of vertebrae and intervertebral discs between C7 and S3), but sampled in 11 points (i = 1, 5, 9, …, 41) to simulate the 11 markers placed on the spine. These 11 points are the input for the spline interpolation and signal-smoothing procedure. The procedure outcome is given by the smoothed helix and its first and second derivatives, evaluated at 41 points, identifying the whole spine. 

The specially developed signal processing procedure has been validated by applying three different simulation tests.

First, to simulate the effect induced by the reconstruction error, a zero-mean white noise e(i) with σ = 0.3 mm (calibration error) has been superimposed to the helixes.

Second, a Montecarlo scenario (3000 repetitions) has been implemented on such noisy helixes to simulate a marker misplacement error mme(i) in two ways:(1)Along longitudinal direction (i.e., below or above the vertebral spinous processes) by adding a white noise mme(i) with σ = 5 mm to z(i), i.e., z(i) = k*i + e(i) + mme(i);(2)In all possible directions, adding the same white noise mme(i) with σ = 5 mm to all three coordinates. Such a test has been applied only to helixes simulating larger spinal deformities (larger Cobb angles) because it has been shown in the literature that this is the most critical condition for marker placement [42].

The amplitude A[j] is computed in metres and is varied following the formula: A[j]: = 0.025 + 0.025*sin(*π**j/10) 0 ≤ j ≤ 10(2)

While the frequency f[j] is varied following
[j]: = 2.0 + 0.50*cos(*π**j/10) 0 ≤ j ≤ 10(3)
where j is the helix index. 

When f[j] = 2, the related sinusoidal helix component completes an entire cycle in the 41 points representing the spine. 

The procedure outcomes are reported in the Results section by comparing the angles computed through the assessed derivatives with the true analytical ones. For the Montecarlo simulations, the mean and standard deviations of the error in the spine deformities assessed angles are reported.

## 3. Results

### 3.1. Error Analysis of 3D Spine Reconstruction Algorithm through Numerical Simulation

The specially developed signal processing procedure results on the above-described simulation tests are reported in Table 1, Table 2 and Table 3. As it can be noted in Table 1, the capability of the algorithm to determine the correct angle value is excellent, when the σ = 0.3 mm calibration noise only is considered, either in the accuracy or in the precision. Indeed, both the mean error and associated standard deviation result to be only a fraction of a degree. The procedure works very well even in the most critical case when the inflexion points are precisely at the signal edges (the angles computed with one inflexion point at one of the borders are highlighted in bold in Table 1). In particular, the worst condition is registered when the values of the first and second derivatives of the actual signal are considerably different from the zero value precisely at its boundaries (i.e., C7 or S3 vertebra). Indeed, the interpolation using natural splines implies setting the value of first and second derivatives at the edges to zero [84], forcing the spine’s derivatives values at the edges. Even under such worst conditions, the maximum error recorded is just 2.71 degrees. This result demonstrates the reliability of the procedure in the 3D spine shape and curvature assessment. Table 2 shows the algorithm’s substantial insensibility to the marker-misplacement error along the longitudinal direction (i.e., simulating a misplacement below or above the vertebral spinous processes), even if the magnitude of such error (σ = 5 mm) is almost 17 times greater than the calibration error. Indeed, the accuracy and the precision are still excellent, having the same magnitude as the previous test. The maximum assessment error is around 3°, still below the typical error ±(3° to 5°) value considered on X-ray measurements [89,90,91,92,93]. Finally, even in the most critical simulation represented in Table 3, the mean error is 2.52°, which is only 4.65% of the true mean angle value.

### 3.2. Clinical Case: Adult Scoliosis Associated with Thoracic and Lumbar Spine Pain and Leg-Length Discrepancy

In the following, we describe a paradigmatic example at length to show the presented multifactorial approach’s capability to process many different measurements, thereby allowing all the results to be combined in a unified view. The evaluation presented here is extracted from an ongoing cross-sectional observational study on low back pain.

The Ethics Committee University of Medical Sciences, Poznan, Poland, approved this study. Resolution number: 376/17. The participant gave written informed consent before the data collection began. 

#### 3.2.1. 3D Posture Static Analysis and Pre–Post Therapy Comparison 

In this example, the patient is a 50-year-old female physiotherapist with a former diagnosis of adolescent idiopathic scoliosis, who needed to momentarily suspend working activity due to thoracic and lumbar spine pain. X-ray and MRI evaluations documented degenerative changes in both the vertebrae (arthrosis) and intervertebral discs, compression to the radicular nerves with osteophytes/bone spurs at the various level, either at the lumbar or thoracic spine. In the frontal X-ray film, the L5 vertebra appears wedged. The patient was coming to our centre for a fully functional evaluation using the GOALS-EGG system. The results of such functional assessment allowed customised therapy planning. Two following sessions were scheduled. As detailed below, the analysis of standing posture in the first evaluation unveiled the patient was presenting a significant leg-length discrepancy (LLD) associated with spinal deformities, postural and underfoot loads asymmetry, excessive pronation of the ankle in the longer limb and conversely, excessive supination of the ankle in the contralateral shorter limb. Such a condition induced incorrect support of the foot either in standing or in gait. This latter analysis confirmed a series of asymmetries, either the step lengths or trunk and global lateral leaning during walking. The derived biomechanical–functional hypothesis was to correct the LLD and the feet’s support using customised insoles. A set of muscle-conditioning therapeutic exercises was planned to stimulate the reduction of postural asymmetries and restore lower-limb functional symmetry during gait. Such a combined therapy plan led to reduced spinal deformities, the complete remission of the patient’s back pain, improved symmetry in the standing posture, and an improved lower limb functional symmetry during gait. The following describes how the GOALS-EGG approach was fundamental in evaluating and constructing the initial hypotheses, identifying individualized therapeutic intervention and verifying/monitoring the intervention results. The first evaluation was during the acute phase and the second one at control after a 4-month rehabilitation period. 

The 31 markers-set protocol described above was used in both measurement sessions. Initially, standing posture characteristics were evaluated in three conditions: IO, SCO and WCO. Figure 3 shows the perfect agreement between the frontal plane projection of stereo-photogrammetric 3D spine reconstruction in IO and X-ray spine deformities measurement (anterior-posterior view; the lateral view is also represented). Figure 4 panels a, b and c show the 3D (averaged) posture and spine shape standard reports of the patient measured in IO, SCO and WCO, respectively. 

The full 3D skeletal posture reconstruction in both the frontal and sagittal plane is depicted. 

The SCO posture measurement accounts for the changes induced by the voluntary performed instinctive self-correction manoeuvre [27] and makes visible how the patient can reduce her global and spinal side unbalancing but at the cost of increasing spinal deformities in the frontal plane. Difficulties are also highlighted in the sagittal kyphosis, where it is evident a block interrupts the curvature continuity in two segments. The underfoot loads’ asymmetry increases from about 2% of body weight in IO to about 9% in SCO, with the load’s prevalence on the longer left leg. In WCO, the LLD has been corrected by placing an underfoot wedge, the optimal value of which (20 mm under right foot) was determined as the one producing the best global posture outcome considering all the combined spine deformities and postural parameters. Indeed, a beneficial global effect on almost all the postural variables (reductions are evident in spinal deformities, global and spinal unbalancing, underfoot load distribution) is visible, except for the block on thoracic kyphosis still present. The evaluation of foot–floor interaction through baropodometric measurements in static posture and during gait suggested using customised foot orthoses to reduce the recorded ankle pronation at the left side and ankle supination on the right side (see Figure 5). The patient was provided with a detailed therapy plan and a pair of customised foot orthoses incorporating the 20 mm heel lift on the right foot. 

After 4 months, the patient returned to control, showing complete remission of back-pain symptoms with self-perceiving enhanced mobility all along the spine. 

Figure 6 shows the cross-session comparison between neutral IO at first evaluation vs. WCO (when the patient donned her foot orthotics) at control. Consistent improvements are evident in each postural variable:Reductions of spinal deformities in the frontal plane at each sacro-lumbar, lumbar and thoracic level;Perfect horizontal realignment of the pelvis;Realignment of feet–pelvis–shoulder axes on the horizontal plane with the complete vanishing of the rotations present at the first evaluation;Underfoot load complete rebalancing.

#### 3.2.2. Spinal Mobility Dynamic Analysis: Lateral and Forward Bending Analysis and SEMG Recording—The Flexion Relaxation Phenomenon

A dynamic flexion analysis verified the recovery of spinal flexibility after treatment. Dynamic flexion analysis was not possible at first evaluation because of the patient’s back pain. Figure 7 shows the comparison of performances between with and without foot orthoses during lateral bending. As expected, the spinal deformities limit side flexions when the movement contrasts to curves’ convexities. In any case, good global mobility is observed. 

Foot orthoses affect pelvis movements and gradually lumbar and upper spinal mobility. SEMG activities on bilateral multifidus and erectors spinae muscles [64] during the forward-bending performance are also reported. A regular patient’s paravertebral muscle activities are recorded, showing the flexion–relaxation phenomenon’s healthy behaviour [66,94,95,96,97]. Some speculation on the dissimilar left and right multifidus activity levels and how foot orthoses influence their contractions can be discussed. Future studies will address this matter. As evident, LLD affects pelvis and upper-body levels not only in static but also in dynamic conditions. LLD effects will be fully explored also in the next gait analysis paragraph. 

#### 3.2.3. 3D Skeleton Gait Analysis on a Baropodometric Treadmill 

Such a paradigmatic example highlights how the GOALS-EGG multifactorial approach allows for correlating standing posture analysis with functional gait evaluation into a unified view and using the results for clinical purposes. This clinical example describes how changes before and after treatment can be studied, documented and validated from a statistical perspective. We focus the analysis on a most representative subset of variables that, highlighting the asymmetries in the biomechanics of the gait, was the support to determine the most effective therapy planning (including information to provide custom-made foot orthoses) prescribed at first evaluation. The first set of figures (Figure 8, Figure 9, Figure 10 and Figure 11) shows the averaged joint angles time courses of mean gait cycles (in various conditions) computed over multiple strides. Figure 8 shows the left vs. right leg comparison during barefoot walking at the first evaluation. IED values at the top of each panel quantify and summarise the percentage of gait asymmetry for each joint angle. The hip flexion–extension angle panel of Figure 8 is zoomed (Figure 9), highlighting how the point-by-point *t*-Test allows for the identification of the sub-regions in which the left vs. right variable’s temporal courses are statistically dissimilar. In such a zoomed panel, it is possible to notice the stance phases sub-events. They are automatically identified by elaborating baropodometric data and segmenting the foot-pressure maps into foot sub-regions (i.e., rearfoot, midfoot, forefoot).

Conversely, Figure 10 describes how the left vs. right leg comparison result changes from barefoot walking at the initial condition to walking with foot orthoses at the control session. As shown, IED values at the top of each panel indicate a general asymmetry reduction passing from the first to the second condition. In two panels, however, i.e., knee flexion/extension and ankle pronation/supination, there is an increase of IED values indicating a more considerable extension in the right knee during the pushing-off phase and a more significant pronation in the right ankle along with the whole stance phase. This latter effect was desired and induced by the customised foot orthosis. Such better left vs. right symmetry achieved in gait with foot orthoses worn is also confirmed by the muscle powers evaluated for each joint of the lower limbs, as evidenced by the IED shown in Figure 11. The whole lower-limbs motor control reorganisation, as stimulated by both the therapy period and the ankle pronation/supination and LLD corrections (induced by custom foot orthoses), is also confirmed by baropodometric analysis (Figure 12). 

Figure 12 shows that passing from barefoot walking in the initial condition (panel a) to walking with worn shoes and foot orthoses at the control session (panel b), the mean gait cycles strides and steps lengths are much more symmetrical. In fact, at control, the left vs. right average stride lengths are identical and the left vs. right average step length difference passes from 3.4 cm to 1.9 cm. In the lower panel of such figure, the peak pressure maps (panels (c,d) and (g,h), respectively) and the differences maps (panels (e,f) and (i,j)) are depicted, respectively. 

The pressure maps (e) and (f) (and analogously (i) and (j)) are obtained by calculating the difference between anatomically corresponding cells’ pressure values. i.e., map (e) is obtained as a subtraction between maps (c,d), while map (f) is obtained as a subtraction between maps (d,c). Therefore, map (e) shows all and only the pressure cells for which the map (c) (left foot) has a value higher than the corresponding pressure cell value in map (d) (right foot); in map (f) the process is reversed. Thus, the two maps (e) and (f) (and analogously (i) and (j)) express the two feet’ different load distributions. The pictures show how the left vs. right load distribution’s asymmetry decreases while the patient walks with donned shoes and foot orthoses. 

Figure 13 shows multiple panels with a column of three data charts representing, from top to bottom, the comparison of the time-course variations along the mean gait cycle (barefoot gait at first evaluation vs. shoes and donned foot orthoses gait at the control) of the following variables: (1) averaged spinal offsets: this variable gives information about the extent and the symmetry of trunk side leaning along the entire cycle; (2) averaged global offsets: this variable gives information about the extent and the symmetry of full-body side-leaning along the entire cycle; (3) the Cobb angle values variation of each spine curve deformity that has been identified in the IO condition; in this last chart, also pelvis frontal plane oscillation (angles variations of the horizontal pelvis axis) is represented. All these variables registered an evident decrease of values at control, indicating reduced and more symmetrical side global and trunk leaning, reduced spinal deformities on the frontal plane and reduced and more symmetrical pelvis horizontal oscillations. Such reduced and more symmetrical lateral inclination and the associated reduced horizontal oscillation of the pelvis induce a substantial reduction (about 20%) in the calculated intervertebral loads (forces and torques) represented in the lowest central panel of Figure 13 during the mean gait cycle. In particular, this panel represents the intervertebral loads calculated at the instant of the mean gait cycle (for both conditions), in which the sum of all intervertebral load values reaches its maximum.

## 4. Discussion and Conclusions

The GOALS-EGG has been built up by merging a stereophotogrammetric system to a baropodometric treadmill, allowing the simultaneous measurement of both kinematic and underfoot pressure distribution maps. The project aimed to overcome the practical and methodological limits of the previous solutions presented in the literature. This was achieved by merging validated models of the lower limbs described in the literature into the 3D parametric biomechanical skeleton model (upper limbs not yet included) developed by D’Amico et al. [17,18,19,24,25,26,27,28,29] and by extending the features of the latter to kinematic and kinetic analyses of gait and run. In the results section, the case study showed how it is possible to evaluate the postural anomalies of upright posture and study how these can induce alterations in dynamic motor tasks such as walking, flexion–extension and trunk lateral bending. The case study also allowed for the showing of the multifactorial approach’s capability to process many different measurements into a unified view, merging information from both the static posture and gait analyses from which a rehabilitation plan was derived. The same multifactorial approach allowed for the capturing of the effects of such a rehabilitation plan and the evaluation of its effectiveness in the trunk, pelvis, lower limbs and foot–floor interaction biomechanics.

Various innovative methodological aspects were introduced. 

To overcome the single-trial measurement studies limitations, the ensemble of performed tasks to enhance the assessment’s statistical reliability must be considered [19,24,25,65]. To this aim, specially developed algorithms and signal-processing procedures were implemented to elaborate all the biomechanical variable of interest (including baropodometry) up to the mean behaviour extraction for both posture and cyclic-repetitive tasks (such as multiple strides in gait and run). The mean behaviour extraction leads to multiple benefits. Indeed, the repeatability and consistency of the assessment can be fully explored using a statistical approach that lends substantiation to clinical significance. This was accomplished by implementing an original method to statistically highlight differences in the mean movement task executed in various conditions (e.g., pre–post treatment, with without orthoses, etc.) or to emphasise side-to-side asymmetries (such as comparing left vs. right behaviour during gait/run). In the Results section, once the normalised mean gait/run cycles were obtained, we applied the point-to-point paired *t*-Tests comparing the congruent variable’s time courses to show both the patient’s lower limb asymmetries and the therapy-induced changes. The newly introduced IED summarising statistical index (such an index was introduced in the literature only for ground reaction forces components [88]) quantifies, in percentage terms, the proportion of the averaged temporal courses of pairs of congruent variables that have statistically different results. Subdividing the gait cycle into sub-phases makes it possible to compute IED expressed relatively to specific phases of the gait cycle (here not presented for brevity).

The soft-tissue artefacts drawbacks affecting the lower limbs gait/run analysis were reduced by developing the optimisation method proposed in the literature [78].

Three other innovative features have been introduced in baropodometric data processing. The original algorithm to determine the baropodometric measure of plantar support while walking on the treadmill, the mean stride computation extended to underfoot pressure measurements and the foot pronation–supination characterisation by the right vs. left pressure map symmetry analysis are all additional innovative features introduced with the presented methodology. 

Various authors tried to integrate baropodometric and kinematic measurements. The first proposal can be found in Giacomozzi et al., 2000 [55], which integrated an opto-electronic system, a force plate and a baropodometric platform, all such devices being fully synchronised by a trigger signal. Unfortunately, this multi-sensor and multi-device system remained only at the prototype and research levels and did not find routine application in the clinical setting. Subsequently, other authors have proposed similar solutions, mainly used for the detailed study of the foot joints’ biomechanics [56,57,58,59,60,61,62,63]. However, these solutions required the need for offline spatial and/or temporal realignment of the measurements via software algorithms.

Conversely, synchronisation in the presented approach is ensured via a hardware timing genlock signal. In this way, it is possible to use baropodometric-embedded measurements for the automated algorithmic-based detection of gait/run-time events. Thus, it is possible to set the calculation of the mean gait cycle reliably and automatically. Thus, operator intervention is needed only to check for source-data integrity.

By segmenting the foot pressure maps into foot sub-regions, even stance phases sub-events (i.e., heel-contact and load acceptance, forefoot contact, rockers and push-off phases) [98] are automatically identified. Such a feature, not applicable when force plates are used, is particularly useful to understand how full-body kinematics evolves in each sub-phase. Worth noting, the biomechanical device market offers the possibility to use treadmills equipped with embedded force platforms. Treadmill vs. overground gait and running has been long compared in the literature. Despite the evidence of some actual kinematic–kinetic motor control differences [99,100] also being correlated to different treadmill mechanical properties compared to the ground [101], their similarity showed promising results for the use of a treadmill in a clinical environment [99,100,102,103,104,105]. In our experimental setup, a kinematic stereophotogrammetric system has been integrated with a specific baropodometric treadmill system. Using this latter device, Reed et al., 2013 [106], demonstrated the repeatability of measurements and the substantial equivalence of temporal, spatial and kinetic gait parameters compared to overground walking. On the other hand, Van Alsenoy et al. [107] established, using the same kind of treadmill, that the reliability was acceptable to excellent to evaluate maximal vertical force, contact time and flight time during running.

The treadmills equipped with force plates can capture the 3D ground reaction forces. Conversely, using baropodometric sources, the vertical component alone of the ground reaction forces can be collected. However, it is beneficial to obtain detailed information about a foot–floor interaction and underfoot pressure distributions in posture and movement disorders. For instance, it is necessary to determine the presence of flat and arched foot, ankle–foot pronation–supination, the complete description of stance and, as mentioned, foot rockers’ sub-phases. Such a fine subdivision of stance phases has been proficiently applied to complement the knowledge necessary to drive the customised orthoses design, as shown in the reported case study. For such a reason, we favoured the baropodometric treadmill, coupled to kinematics, at the detriment of the loss of information regarding the transverse components of the ground reaction forces. 

The paper’s further innovative methodological aspect explores the 3D spine biomechanical morphology and functional behaviour either in static or dynamic conditions.

The presented methodology shows significant benefits in the analysis of spine morphological characteristics and disorders. Several attempts in recent years have been made to develop non-invasive methods to resolve 2D single-shot X-ray limitations. Alas, many of these modern methods still reflect certain restrictions. In some instances, they require the subject to maintain a particular and restricted posture that significantly affects the outcomes, as is the case for low-radiation dose 3D X-ray devices [12], yet do not allow a complete posture and spine functional evaluation. In other cases, in addition to restricted posture (rasterstereographic back-surface measurement [16,20,21,22,23], the recently introduced ultrasound approach [16] or the less sophisticated electro-goniometric and/or flexicurve devices [1,20,108]), there are concerns and questions about measurement accuracy which need further clarification. 

In spine disorders, limitations in functional analysis lead to incomplete evaluations and therapeutic planning. For example, in ASD, the literature describes a treatment approach focused on a surgical sagittal/frontal deformities correction and successively evaluated the postoperative-induced changes in global posture and lower-limb compensatory mechanisms [13]. Such a descending-path approach, i.e., correcting the upper part and evaluating the outcomes down to the lower limbs, disregards possible active influences on the spine deriving from lower-limbs posture corrections, possibly in a conservative rehabilitation approach. Worth noting, the surgical approach is hazardous for older patients, with a high rate of postoperative complications (37% to 62%), requiring lengthy recovery times along with being very expensive [109]. Furthermore, the success rate is relatively low. After surgical correction, in ASD, low congruity of the postoperative alignment with age-adjusted ideals has been observed, with less than 30% of patients matching age targets spinopelvic parameters [13].

Maybe such a low success rate for surgery could be related to the unavailability of systems that permit a quantitative functional test to derive pre-surgery patients’ complete postural characteristics.

Optoelectronic stereophotogrammetry provides an unobtrusive and harmless yet complete significant solution for capturing the full-skeletal biomechanical characteristics. However, this approach’s main difficulty was to obtain a kinematic model of the spine, seen as a continuous deformable body yet providing enough detail to consider the position of the single-vertebral body. Moreover, it is desirable to obtain adequate spine motion measurements while keeping a small set of surface markers.

Some literature proposal modelled the spine in various rigid segments (including multiple vertebrae) identified by clusters of markers [39,40,41] (see Needham et al., 2016 for a review [47]). In others, the vertebral column was modelled by linear segments joining markers placed one every two spinous processes [31,32,33]. Finally, more sophisticated models used markers positions (generally placed on the spinous processes of one out of two with few exceptions) as the basis for a polynomial interpolation [34,35,36,37,38] or fitting circumferences [42,43,44] to reconstruct the 3D spine shape and assess its curvatures.

Apart from the chosen mathematical model, an additional concern is about marker placement. Marker positioning has been historically considered the most critical concern due to the inappropriate belief that the manual cutaneous placement of the passive retro-reflective markers on the palpable bony landmarks inevitably leads to inaccuracy results, in particular at the spinal level. Studies based on the comparison between stereo-photogrammetric vs. X-ray or MRI measurements confirmed that trained operators could perform correct marker placement via palpation on predefined bony prominences, even on the spine. These studies also quantified the error due to incorrect marker placement [30,36,42,110,111], demonstrating the weak influence of such misplacement error on the spine curvatures assessment, being the angles in the sagittal plane estimated with reasonable accuracy through stereo-photogrammetry. About markers placement repeatability, it has been assessed that the subject spine’s postural variability was far greater than the intra- and inter-examiner error [112,113]. Schmidt et al., 2015 [42], analysing a group of scoliotic patients (mean X-ray-measured Cobb angle of 44.4°), observed that, in the frontal plane, skin markers curvature assessed angles could be underestimated if severe scoliotic curves are present. This underestimation may be partly due to marker placement difficulty on such patients and the intrinsic systematic axial rotation of the vertebrae induced by scoliotic deformity, which shifts the spinous processes position towards the concavity of the curve [114].

Nonetheless, Severijns et al., 2020 [38] demonstrated that the polynomial method, with subject-specific anatomy correction (requiring a preliminary biplanar X-ray scan with placed radio-opaque skin markers), can measure spinal deformities validly and reliably using motion capture also in subject with severe scoliotic curves. Except for this last example, the polynomial-based methods are mostly limited to describing the spine shape on the sagittal plane. Ranavolo et al., 2013 [36] noted the limiting 2D nature of the information extracted by polynomial interpolation: ‘indeed, despite applying the same interpolation for both the sagittal and frontal planes, the method provides a different curve for each plane (and not a 3D curve)’.

In the present paper, to overcome such a limitation, the method introduced by D’Amico et al. [17,19,24,25,26,48,49,67] was considered using spline interpolation [30] followed by automatic adaptive digital filtering. The algorithm allows the reconstruction of the full 3D spine shape considering all the vertebrae and intervertebral discs positions and attitudes from C7 down to S3. The validity of such 3D reconstruction for static posture has been fully explored in D’Amico et al., 2017 [24] and Kinel et al., 2018 [26], where the normative 3D spine morphology characteristics evaluated using stereo-photogrammetry in a healthy young population confirmed to be consistent with X-ray literature findings. Furthermore, a set of noisy synthetic helixes modelling spine deformations during movement has been tested to extend the signal processing procedure’s numerical validation to the dynamic case. We tested the algorithm’s performances in two ways: considering the measurement error (σ = 0.3 mm calibration white noise) and then performing a Montecarlo simulation (3000 repetitions) by adding to such noisy helixes also a random misplacement error (σ = 5 mm). As shown in Table 1 (measurement error only), the curvature angle assessment error is a fraction of a degree in most cases (mean error = 0.65°). The procedure also demonstrated to work reasonably well in the worst conditions when the inflexion points are found at ‘noisy spine’ boundaries (i.e., C7 or S3 vertebrae). That is when the values of the first and second derivatives of the ‘true spine’ are considerably different from zero in such positions as imposed by the natural cubic spline interpolation [84]. Worth noting, the largest experimental error on synthetic signals in such particularly poor conditions resulted in 2.71°.

Interestingly, the algorithm showed a substantial insensibility to the marker misplacement error along the longitudinal direction. The resulting mean estimation error presented the same magnitude (0.68°) as the previous test, even if the considered misplacement error (σ = 5 mm) is almost 17 times greater than the calibration error. Finally, even in the most critical conditions, simulating severe scoliotic curves and a 3D misplacement error, the mean error resulted in 2.52° (which is only 4.65% of the true mean angle value), while the mean maximum error was equal to 6.26°. All such errors are below the typical error ±(3° to 5°) value considered on X-ray measurements [89,90,91,92,93]. Even when sophisticated elaboration software was used, the radiographic error resulted in ±4°–5° for the Cobb frontal plane angles and kypho-lordotic angles, but when severe scoliotic deformities are considered, it could increase up to ±8° [115]. 

Various authors validated the reliability and repeatability of the spine shape modelling during movement, considering skin surface markers and soft tissue artefacts. This artefact has been shown to introduce a major error during axial rotation when calculating spine motion (14–16 mm during 35° rotation) [116]. The measurement error associated with mild static flexion for the spine markers was also calculated to be about 9–10 mm [111]; nevertheless, the same authors observed that the relative changes in the spine markers positions were a valid representation of the spinal sagittal motion as also found in Ignasiak et al. [37]. Moreover, motion measurement results based on skin markers were consistent with radiographic findings in the frontal plane [117,118], and the application of motion-capture technology for tracking the spinal motion in both normal–healthy and deformed spines has been advocated [119]. 

Nevertheless, the introduced methodology suffers from some limitation.

Undoubtedly the soft tissue artefacts constitute a significant drawback in the 3D spine morphology assessment during movement, thus negatively influencing the intervertebral net forces and torques evaluation. Furthermore, it is essential to consider that the regression parameters provided by Liu and Wickstrom [72] may not be appropriate when trunks with severe spinal deformities are considered. A further limitation of the presented skeletal model relates to the current lack of modelling of the upper limbs, and therefore, the contributions due to the forces and net torques to the shoulder joint cannot be calculated.

Using a protocol that considers a complete skeleton model inevitably involves spending some time on the markers’ positioning. However, such effort is amply rewarded by the possibility of having the complete skeleton posture analysis, quantitative evaluation of the morphology of the spine and biomechanical characteristics of the complete skeleton during walking in a statistically valid framework. 

As explained, by elaborating the synched baropodometric data, the processing up to the computation of the mean behaviours is automatic, fast and requires minimal operator intervention. Therefore, the entire measurement session, including posture and gait analysis, takes less than an hour from patient entry to full-report presentation. Thus, the above appears to be an acceptable trade-off between the achieved benefits and the marker-placement-procedure burden.

In conclusion, such extensive and detailed biomechanical analysis showed the potential to be applicable at the diagnostic stage and in the designing and developing of treatment programs and monitoring the progression of pathology and/or treatment outcomes. The newly introduced methodology extends the power of full-body 3D posture examination by including a fully automatic and statistical framework for a comprehensive movement analysis. This approach can be applied to any postural/motor disorder and pathology, including the orthopaedic, rehabilitative, and neurological sectors; the prevention of sports injuries and the optimisation of performance as well as the monitoring of postural/motor changes due to growth and/or ageing, where the harmfulness of the approach and the easy-to-perform analysis are essential aspects. These features enable the clinician/researcher to perform multiple postural/movement tests per patient and compare them directly in a statistically reliable framework. 

## Figures and Tables

**Figure 1 sensors-21-03930-f001:**
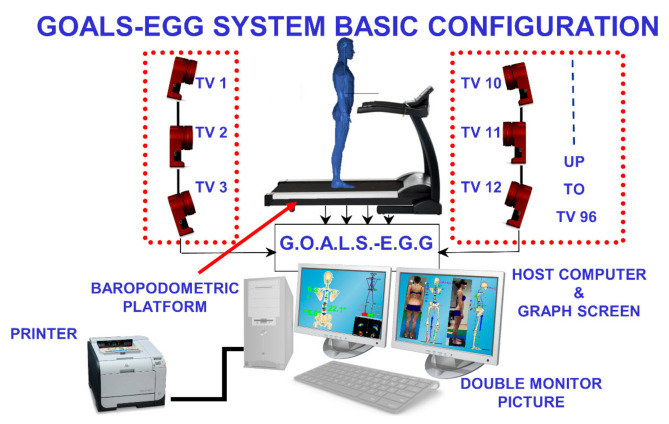
A general schematic configuration of the GOALS-EGG system.

**Figure 2 sensors-21-03930-f002:**
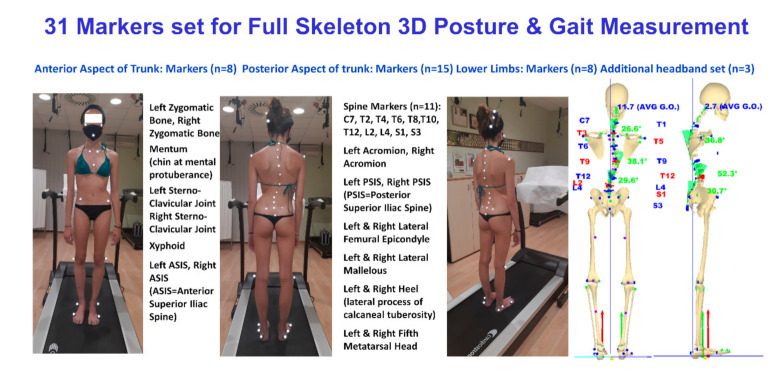
Simplified protocol for 3D posture, gait and run analysis: a list of 31 anatomical landmarks identified by palpation plus a three-marker headband set.

**Figure 3 sensors-21-03930-f003:**
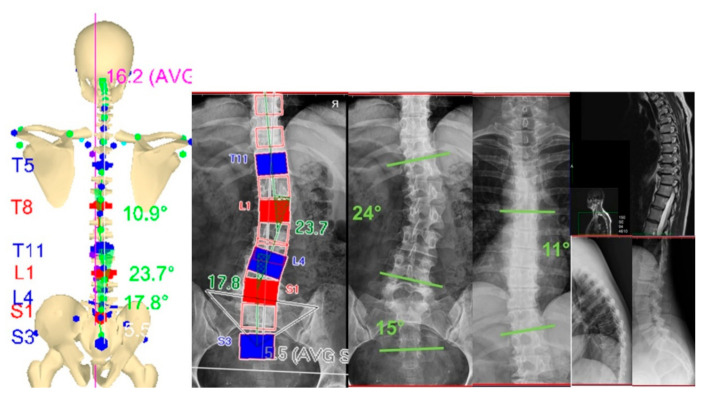
Comparison between stereo-photogrammetric 3D spine reconstruction and X-ray spine deformities measurement in the frontal plane. On the right side, X-ray and MRI evaluations document degenerative changes in both the vertebrae (arthrosis) and intervertebral discs, compression to the radicular nerves with osteophytes/bone spurs at the various level either at the lumbar or thoracic spine.

**Figure 4 sensors-21-03930-f004:**
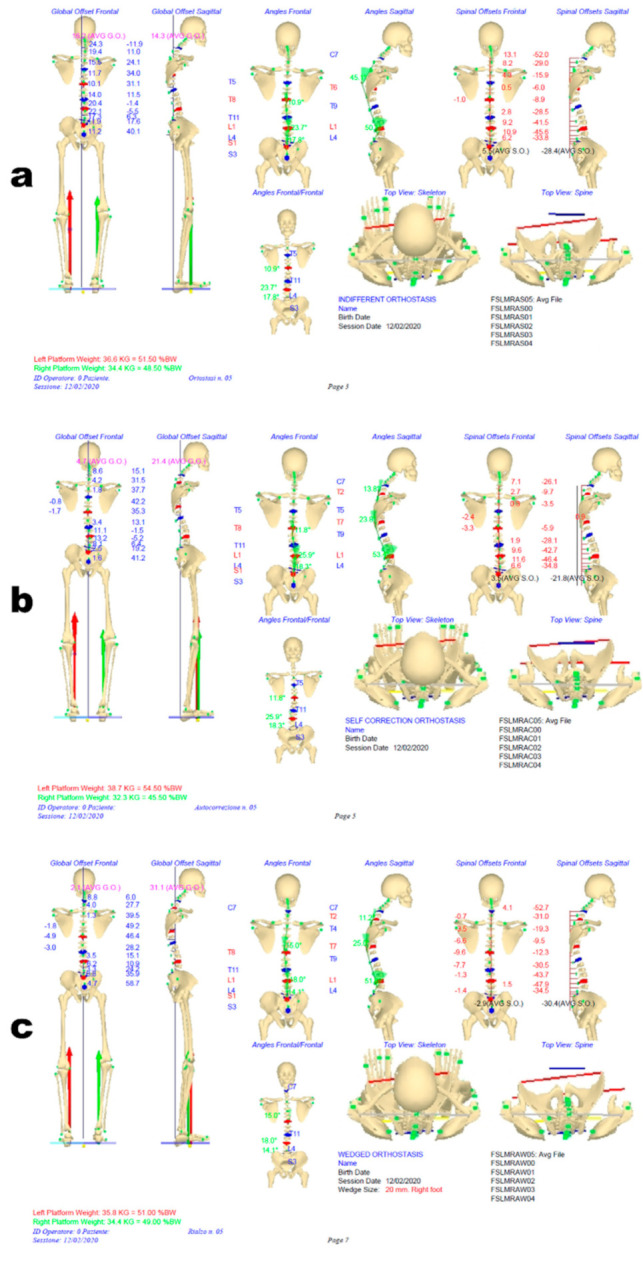
Standard biomechanical reports of the patient’s 3D (averaged) posture and spine shape measured in three different standing conditions: IO (panel **a**), SCO (panel **b**) and WCO (panel **c**), respectively. The end and apical vertebrae (depicted in blue and red colours respectively), Cobb, kyphosis and lordosis angle values, and the spinal and global offset values and their averages, are automatically identified and computed (the horizontal distances in the frontal and sagittal planes of each labelled spine landmark respect to the vertical axis passing by S3 are defined as spinal offsets, while the horizontal distances in the frontal and sagittal planes of each labelled spine landmark respect to the vertical axis passing through the middle point between heels are defined as global offsets; such values are averaged to compute the average spinal and average global offsets as summarising value; negative values represent offsets on the left side in the frontal plane and forward offsets in the sagittal plane). A leg-length discrepancy (being the left leg longer than the right one) was determined in IO. It was associated with spine deformities and postural unbalancing with a lateral-leaning posture on the right side either globally or in the trunk. In WCO, it is possible to evaluate the effects induced by the LLD compensation through an underfoot wedge of 20 mm. The pelvis horizontal alignment was restored, global and spinal offsets reduced to almost zero value and spinal deformities strongly reduced in value.

**Figure 5 sensors-21-03930-f005:**
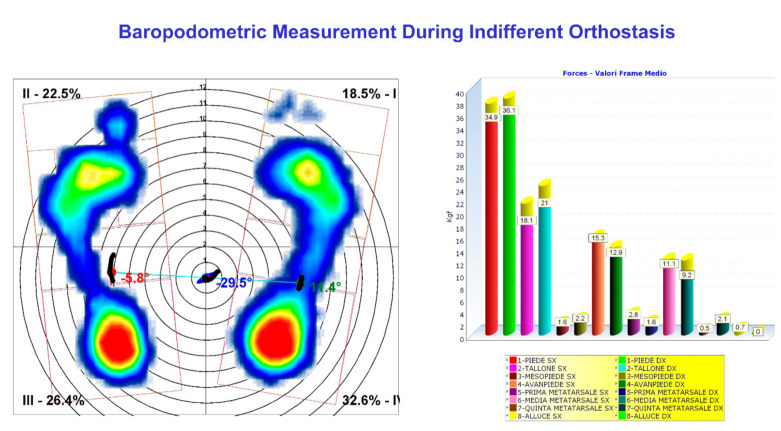
Baropodometric measurement during indifferent orthostasis.

**Figure 6 sensors-21-03930-f006:**
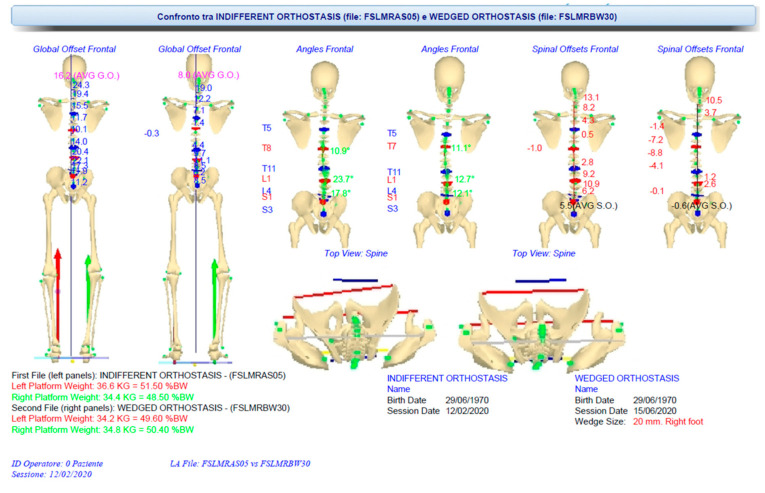
Comparison between IO at first evaluation vs. WCO at control after a 4-month rehabilitation period. (See text and Figure 4 caption for the definition of spinal and global offsets.)

**Figure 7 sensors-21-03930-f007:**
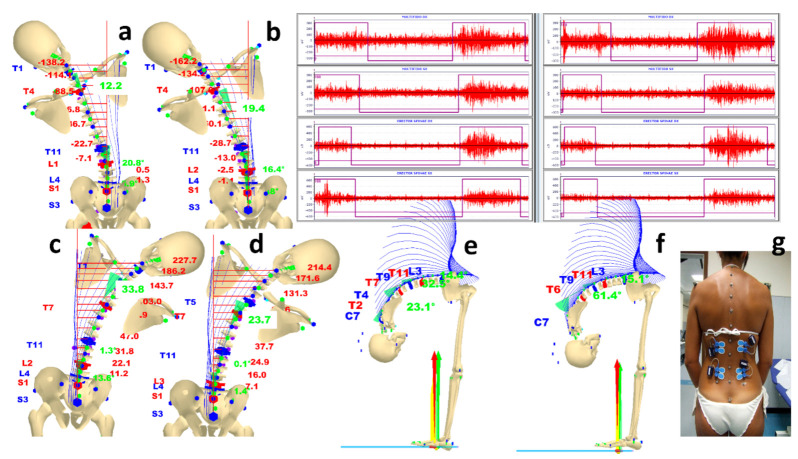
Spine mobility evaluation in lateral- and forward-bending. Panels (**a**) vs. (**b**) and (**c**) vs. (**d**) show the comparison of performances between without (**a**,**c**) and with (**b**,**d**) foot orthoses during lateral bending. Panels (**e**) vs. (**f**) show the comparison of performances between without and with foot orthoses during the forward-bending associated with corresponding SEMG activity. Panel (**g**) shows an example of bilateral multifidus and erector spinae electrodes positioning following SENIAM recommendations.

**Figure 8 sensors-21-03930-f008:**
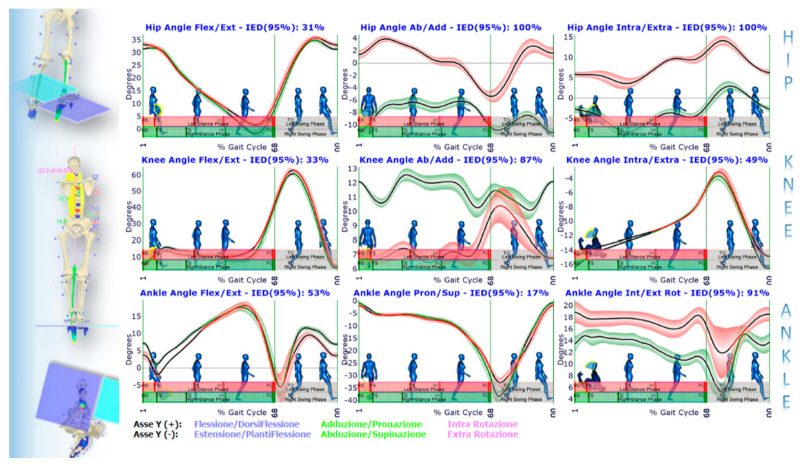
Mean gait cycle: lower limb joint angles. The left (red colour) vs. right leg (green colour) comparison during barefoot walking at the first evaluation is represented. IED values at the top of each panel quantify and summarise the percentage of gait asymmetry per each angle at each joint level. The black coloured lines sub-regions highlight where the left (red colours) vs. right (green colours) variable’s temporal courses point-by-point *t*-Test is significant (*p* < 0.05).

**Figure 9 sensors-21-03930-f009:**
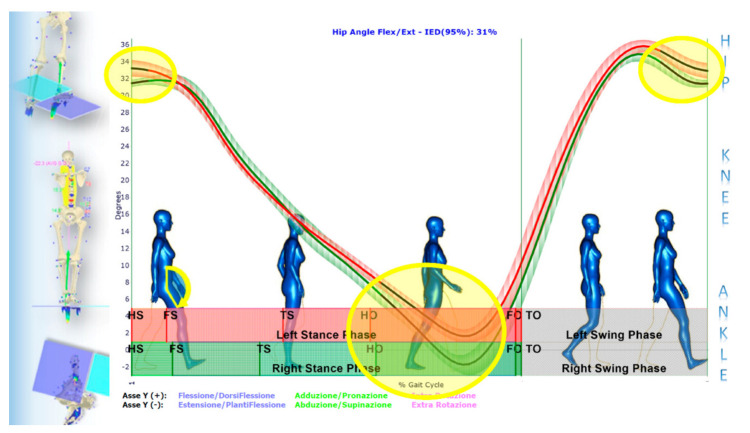
Mean gait cycle: lower limbs hip flexion–extension joint angles. The yellow ellipses highlight the sub-regions in which the left (red colours) vs. right (green colours) variable’s temporal courses point-by-point t-Test is significant (*p* < 0.05) (black-coloured lines). The lower bars, left (red colours) vs. right (green colours), report the stance phases’ sub-events. They are automatically identified by elaborating baropodometric data and segmenting the foot pressure maps into foot sub-regions (HS = Heel Strike, FS = Forefoot Strike, TS = Toe Strike, HO = Heel Off, FO = Forefoot Off, TO = Toe Off). IED values at the top of each panel indicate the left vs. right percentage asymmetry.

**Figure 10 sensors-21-03930-f010:**
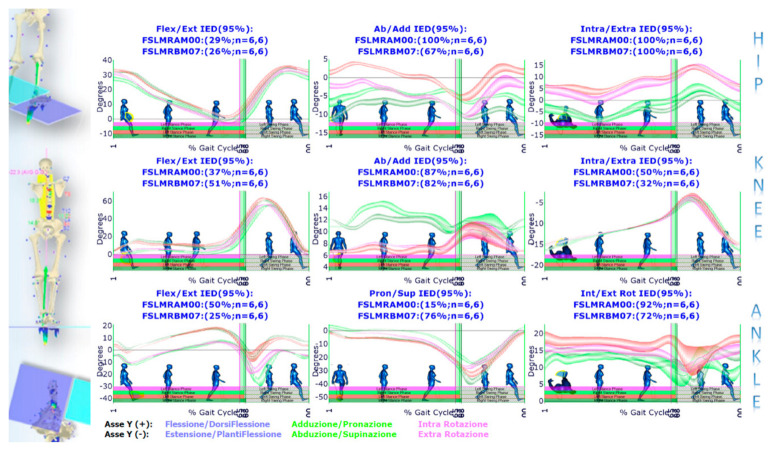
Average gait cycle: joint angles of the lower limbs. Each panel shows the left (red colours) vs. right leg (green colours) comparison in the following conditions: barefoot walking in the initial evaluation (darker colours, file name code FSLMRAM00) and walking with foot orthosis during the control session (lighter colours, file name code FSLMRAM07). The following information is listed in brackets: (1) IED values; (2) the number of valid strides included in the averaging process. From the IED analysis, it is possible to notice a reduction of the general asymmetry passing from the first to the second condition.

**Figure 11 sensors-21-03930-f011:**
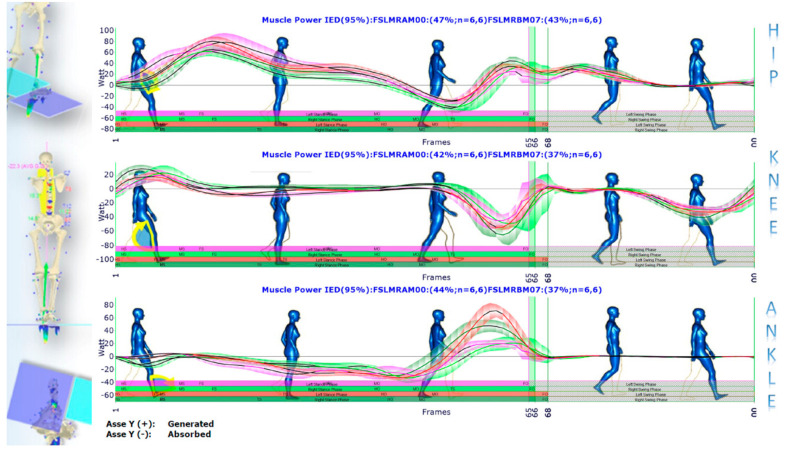
Average gait cycle: muscle power at lower limbs joints. Each panel shows the left (red colours) vs. right leg (green colours) comparison in the following conditions: barefoot walking in the initial evaluation (darker colours, file name code FSLMRAM00) and walking with foot orthosis during the control session (lighter colours, file name code FSLMRAM07). The following information is listed in brackets: (1) IED values; (2) the number of valid strides included in the averaging process. From the IED analysis, it is possible to appreciate the general asymmetry reduction passing from the first to the second condition.

**Figure 12 sensors-21-03930-f012:**
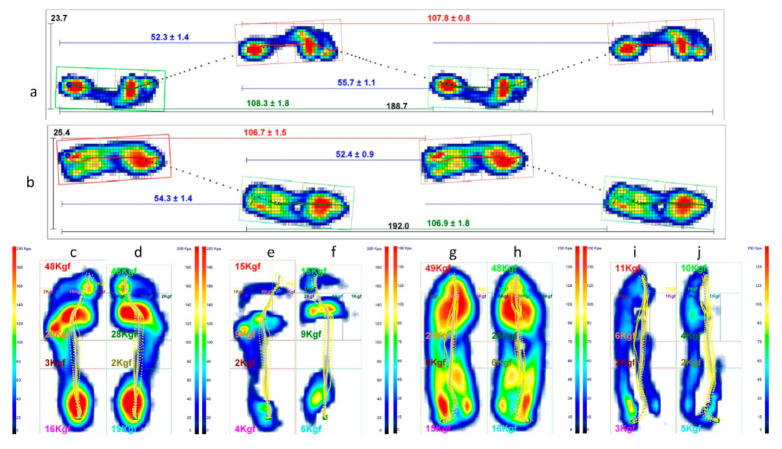
Mean gait cycle: baropodometric analysis (see text for explanation).

**Figure 13 sensors-21-03930-f013:**
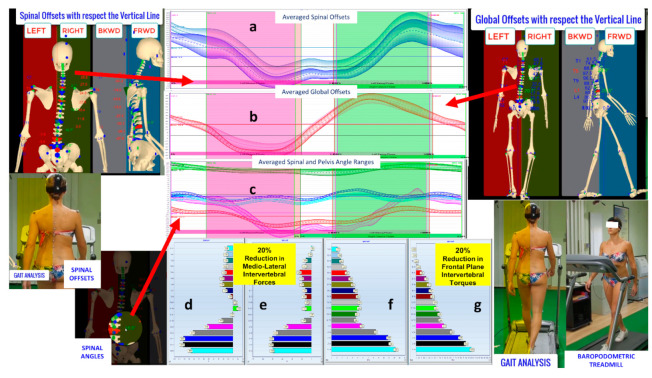
Mean gait cycle. Comparison between barefoot gait at first evaluation vs. shoes and donned foot orthoses gait at the control. Panels (**a**,**b**): comparison of time courses of spinal and global leaning. Panel (**c**): Cobb angle values variation of each spine curve deformity and pelvis inclination (see text for explanation). Lowest central panels: comparison between the instantaneous mediolateral intervertebral forces (panel (**d**)—first evaluation, panel (**e**)—control) and frontal plane intervertebral torques (panel (**f**)—first evaluation, panel (**g**)—control) both taken at the moment in which the sum of all intervertebral load values reaches its maximum.

**Table 1 sensors-21-03930-t001:** Assessed Absolute Angle Error by the specially developed signal-processing procedure on a set of noisy synthetic helixes modelling spine deformations during movement. (The angles computed with one inflexion point at one of the borders are highlighted in bold).

Helix	True Angle	Estimated Angle	Absolute Angle Error
1 (A = 2.5, f = 2.5)	42.47	42.48	0.02
1	42.81	43.22	0.42
**1**	**21.43**	**18.72**	**2.71**
2 (A = 3.27, f = 2.47)	53.46	53.01	0.44
2	53.87	54.30	0.43
**2**	**24.73**	**23.13**	**1.60**
3 (A = 3.96, f = 2.40)	61.47	61.26	0.21
3	61.90	62.16	0.26
**3**	**20.90**	**19.81**	**1.08**
4 (A = 4.52, f = 2.29)	65.78	65.18	0.60
4	66.16	65.75	0.41
**4**	**11.61**	**9.80**	**1.81**
5 (A = 4.87, f = 2.15)	66.51	66.48	0.03
5	66.87	67.33	0.46
6 (A = 5.0, f = 2.0)	63.90	64.21	0.31
6	64.21	65.06	0.85
7 (A = 4.87, f = 1.84)	58.73	60.04	1.31
7	56.06	56.72	0.67
8 (A = 4.52, f = 1.70)	51.54	52.19	0.65
8	42.17	42.01	0.15
9 (A = 3.96, f = 1.59)	43.24	43.28	0.04
9	28.39	28.11	0.28
10 (A = 3.27, f = 1.52)	34.70	34.54	0.15
**10**	**18.78**	**19.83**	**1.05**
11 (A = 2.5, f = 1.5)	26.42	26.21	0.21
11	13.25	14.12	0.88
**Mean Value**	**44.67**	**44.58**	**0.65**
**Standard Deviation**	**18.67**	**19.07**	**0.63**

**Table 2 sensors-21-03930-t002:** Montecarlo scenario (3000 repetitions) helixes modelling spine deformations during movement: Assessed Absolute Mean Angle Error and SD simulating marker-misplacement error along the longitudinal direction.

Helix	True Angle	Mean of Estimated Angles	Absolute Mean Angle Error	Angle Error SD
1 (A = 2.5, f = 2.5)	42.47	42.27	0.20	0.45
1	42.81	43.57	0.77	0.58
1	21.43	20.17	1.25	0.30
2 (A = 3.27, f = 2.47)	53.46	53.11	0.35	1.67
2	53.87	54.75	0.88	1.70
2	24.73	22.97	1.76	0.88
3 (A = 3.96, f = 2.40)	61.47	61.29	0.18	0.47
3	61.90	62.35	0.45	0.66
3	20.90	20.65	0.25	0.20
4 (A = 4.52, f = 2.29)	65.78	65.06	0.73	0.32
4	66.16	65.78	0.38	0.86
4	11.61	11.16	0.45	0.60
5 (A = 4.87, f = 2.15)	66.51	65.34	1.17	0.39
5	66.87	65.88	0.99	0.56
6 (A = 5.0, f = 2.0)	63.90	63.54	0.36	0.40
6	64.21	64.10	0.11	0.49
7 (A = 4.87, f = 1.84)	58.73	58.87	0.15	0.55
7	56.06	55.55	0.51	1.46
8 (A = 4.52, f = 1.70)	51.54	51.40	0.14	0.31
8	42.17	41.59	0.57	0.70
9 (A = 3.96, f = 1.59)	43.24	42.86	0.38	0.44
9	28.39	27.73	0.66	0.40
10 (A = 3.27, f = 1.52)	34.70	34.49	0.20	0.92
10	18.78	17.95	0.83	0.98
11 (A = 2.5, f = 1.5)	26.42	23.37	3.05	3.38
11	13.25	12.45	0.80	1.65
**Mean Value**	**44.67**	**44.16**	**0.68**	**-**
**Standard Deviation**	**18.67**	**18.95**	**0.63**	**-**

**Table 3 sensors-21-03930-t003:** Montecarlo scenario (3000 repetitions) on helixes modelling severe spinal deformities (larger Cobb angles) during movement. Assessed Absolute Mean Angle Error and SD simulating 3D marker-misplacement error (i.e., in all possible directions).

Helix	True Angle	Mean of Estimated Angles	Absolute Mean Angle Error	Angle Error SD
2 (A = 3.27, f = 2.47)	53.46	54.24	0.79	16.09
2	53.87	60.13	6.26	14.01
2	24.73	24.61	0.12	12.44
3 (A = 3.96, f = 2.40)	61.47	60.98	0.49	16.85
3	61.90	66.10	4.21	15.54
3	20.90	21.58	0.68	11.65
4 (A = 4.52, f = 2.29)	65.78	66.84	1.06	15.01
4	66.16	70.11	3.95	13.56
6 (A = 5.0, f = 2.0)	63.90	64.11	0.20	15.98
6	64.21	69.49	5.28	9.95
7 (A = 4.87, f = 1.84)	58.73	62.81	4.08	12.41
7	56.06	59.20	3.14	13.06
**Mean Value**	**54.26**	**56.68**	**2.52**	**-**
**Standard Deviation**	**15.33**	**16.33**	**2.20**	**-**

## Data Availability

The data presented in this study are available on request from the corresponding author. The data are not publicly available due to privacy and ethical reasons.

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
