# Peer review of "A Self-Contained 3D Biomechanical Analysis Lab for Complete Automatic Spine and Full Skeleton Assessment of Posture, Gait and Run"

_sensors, 2021, doi:10.3390/s21113930_

Round 1
Reviewer 1 Report
I noticed this is a revised manuscript and I value the authors efforts to answer all the previous concerns and feel that the manuscript overall improved. It seems to me that most of the previous concerns were well addressed in the revised version. Overall, the revised paper quality largely meets the requirements of "Sensors", hence the manuscript could be accepted for publication after proofreading.
Reviewer 2 Report
Methodology for 3D postural analysis during locomotion seems to be the main theme of the current study. Utilisation of force-plate embedded treadmill with 3D motion capture systems should be the reasonable setup for gait and posture analysis. However, there are a lot of issues in the current manuscript. First, manuscript structure is very odd. Methodology section does not operationally describe all the information necessary for scientific research. Please be reminded that research should be repeatable. Second, I am not very sure what is new in the study. I think the computer interface based on the automated data programming may be an interesting idea to provide easier gait analysis environment. However, I am not too sure what the focus of the manuscript is. Third, readability must be improved. Not only grammatical errors but also there are so many expressions that are difficult to follow. Fourth, results section contains a lot of information but many parameters are not properly defined in methods and I cannot find statistical evidence supporting the obtained results. Further comments are provided below.
General
- Try to follow the usual research paper structure. For example, Method section does not have any information about participants. Experimental protocols are not described well. Marker setup is also vague. Statistical analysis is difficult to understand, which does not seem to be utilised in Results. Please work on ‘structure’.
- Improvement of readability must be considered. Professional assistance may be required.
- Try to tell readers what the manuscript is about. At the moment, I cannot see the straightforward message from the manuscript. Rather, it looks like bits and pieces from the experiment, which has not been properly described.
Specific
- Abstract: Abstract is too long and too vague. Fundamental revision must be required to accommodate the journal's guideline. I believe abstract should be 200 words or less, structured without headings and a single paragraph.
- Line 68: ‘…showed to provide…’ Isn’t is supposed to be ‘has been shown to provide…’?
- Line 82: ‘…the sagittal projection of the spine’ Why only sagittal? If it is the 3D measurement, doesn't it affect all axes equally?
- Line 90-101: It is difficult to see the link from the previous paragraph. Out of nowhere, foot pressure measurement emerged. Without providing a proper linkage, it is hard to understand how foot pressure and spine alignment can be related. Some grammatical errors are observed (e.g. '...the primary focus has mostly devoted to...')
- Line 102-106: Please improve the readability here. This seems to be the main research topic of the current study. It is probably better to briefly explain what the 'close connection' is between the spine and baropodometry, first. Then, this research aim can be justified better and readers can follow this easily.
- Materials and Methods: Do you have any information about participants?
- Line 151: What is FDM? Are you talking about dual-force plate embedded treadmills? If so, how were the force plates split?
- Line 258-259: What do you mean 'this latter'?
- Line 271 onwards: Frontal plane and coronal plane should be the same thing. Sagittal plane also has vertical coordinates of PSIS.
- Line 277: Double-check the expression Forward direction can be defined only by Y axis. If forward direction has to be defined, it should be anterior direction but not anterior-posterior.
- Line 278: All planes are orthogonal to each other.
- Line 296: Check grammar. ‘with respect to’??
- Line 312: Sternum is in the front part of the body whereas thorax sagittal angle can be measured based on the backbone. Please double-check if this is correct.
- Line 322: Please describe statistical tests used in the study more properly.
- Line 326-327: Indicate walking speeds.
- Results: In general, the section seems to present interesting information. However, I am not sure if this presentation is understandable. Are these results based on the experiment? How many participants were involved? Are all testing procedures properly described in Method section? The author mentioned about statistical approach but Results were not really supported by statistics. Although there are probably good contents here, it is strongly suggested that the manuscript is structured in the proper format for a scientific paper.
Reviewer 3 Report
This study is introducing new developments and integrations in previously presented an Opto-electronic Approach for Locomotion and Spine system. The developments are including integration of baropodometric system and complete biomechanical skeleton model. The text is well described and well documented and the whole perspective could be the interests of readers.
However, there are several comments remains to be discussed:
- The main question is what is the scientific and research question? In this matter, authors proposed very less materials apart from a new development in signal processing. All these system, models and methods have been introduced previously and now just have been presented in a commercial pack again. There is no comparison between the results which is coming from only one subject by other methods or models. Nothing to imply why a researcher or clinican should prefer this system rather than others (again from scientific and research point of view).
- There are several claims presented here without sufficient evidence and results to support which I mention some of them
- “the GOALS-EGG has been built around the complete 3D parametric biomechanical skeleton model…” in this study no evidence has been presented to show the spine or upper limb in a biomechanical model concept. At very first when we talk about a model, the segment definition (proximal and distal ends) should be exactly determined. What we have here is a measurement method.
- “By integrating baropodometric data, the model allows the estimation of internal lower limb and trunk joint forces, torques, and muscle efforts through data fusion and optimization procedures.” No results have been presented to support muscle efforts. It is not clear what is trunk segment, what is the trunk joint and consequently what is the trunk joint force. Where are the results?
- “In this way, even stance phases sub-events (i.e. heel-contact and load acceptance, forefoot contact, rockers and push-off phases) are automatically identified.” This is an interesting claim but again no data to support and I’m personally believe the system may fail for patients with foot deformities (i.e. equines foot, crouch knee).
- There are some technical issues that should be mentioned too.
- Authors mention about defining new features as spinal and global offsets using midpoint between heels. Two feet despite of hip joints (linked with pelvis) are independent and yet there is not exact definition of placing heel markers. Either authors could present their method about how they aligned the feet and establish a robust heels axis which could be very interesting or this axis leads to poor outcomes (i.e. in patient with foot deformities).
- “in a right-handed system: the frontal-coronal (YZ) plane is the vertical plane containing the PSIS.” This definition is not clear. Do you mean YZ plane passing PSIS and S1? In this case (also according to your figure) the three markers are almost aligned horizontally and produce errors.
- Figures quality in general are very low. No a,b,c are deifned in figure 4 caption. IEDs could not be read in graphs.
Round 2
Reviewer 2 Report
Answering the general response from the authors, I have not yet found anything really new in terms of technology.
‘We think that there has been a basic misunderstanding in that the paper aims to present a new multisensor instrumental and methodological approach to integrating baropodometric information with the entire skeleton's kinematic data, including the 3D reconstruction of the spine, to evaluate either static postures or the thorough description of the movement of the complete skeleton during gait or different functional tasks.’
Precise motion capture system (i.e. Optotrak, Vicon), Force-plate embedded treadmills (e.g. AMTI), EMG and foot-pressure measurement insoles are all available and often used together for data collection.
The only new things seem to be; (1) the data processing algorithm including software interface and (2) automatic synchronisation of the multiple measurements.
These things are important to advance biomechanical research in general.
However, writing scripts by itself may not be considered something innovative unless the details have been described in the manuscript.
Automatic synchronisation of multiple measurements may have some merits in terms of accuracy. However, the system’s accuracy may not be as good as other standard measurement equipment. The developed system should be tested based on the large gait data.
Please try to highlight what is new in the current technology. At the moment, it is not convincing whether this proposed system should be widely used in clinical settings. For example, clinicians usually find it difficult to use a lot of markers but the current system seems to require a relatively comprehensive marker setup, assuming excessive preparation time.
Many gait analysts utilising force-plate embedded treadmills may have their own algorithms for data-processing and I do not understand whether the research adds any new information to data collection for gait biomechanics.
Gait analysis system all synchronised together may be somewhat new but may not be practical due to complicated and time-taking marker setup. Clarify why clinicians prefer this system over all the other equipment already available.
In the reply, the authors repeatedly mentioned that the reviewer has misunderstood a lot of information in the manuscript. It is, however, the authors’ responsibility to avoid misinterpretation and as advised previously, the manuscript is difficult to follow due to many language issues and the professional assistance should be considered for readers to understand the main contents. Comments are provided below.
- Abstract: Previously, I mentioned that the journal accepts the maximum of 200 words, which is just in the generic 'author guideline'. The authors said that they shortened the abstract to less than 300 words but I wonder if they have the intention not to comply with the journal’s guideline.
- Line 25-27: ‘…a complete 3D parametric biomechanical skeleton model, developed in an original way for static 3D Posture analysis, to kinematic and kinetic analysis of movement, gait and run.’ What kind of kinetic analysis is possible from the vertical ground reaction force data and kinematics?
- Line 117-119: Please refer to the comment 2.
- English language: Readability of the manuscript should be improved to the sufficient standard. The authors are responsible for readers not to misunderstand the contents. Professional writing assistance may be required.
- Line 622-635: This section seems to be the justification for not recording horizontal GRF components but the use of foot-pressure measurement insoles such as F-scan is not a difficult option.
Reviewer 3 Report
Thanks to authors for their efforts to answer my concerns.
I have no more questions.
Author Response
Thank you for your review.
This manuscript is a resubmission of an earlier submission. The following is a list of the peer review reports and author responses from that submission.
Round 1
Reviewer 1 Report
This paper aims to define the current level of progress of Opto-Electronic Stereo-photogrammetric approach as a clinical tool and its versatility. Generally, the paper is interesting to my mind, however, there are changes that could be made to make the results more accessible and clear to readers, in details:
I am concerned about the skeleton mode, it seems that fine movement can not be identified using the proposed skeleton mode as shown in FIGURE 2.
In section 3., the authors give some figures and tables. However, they do not make strong conclusions. Furthermore, more validation should be given.
The manuscript is not focused enough, both abstract and conclusion section are too lengthy, the authors should clarify what is important, what is new, and how this proposal is advancing knowledge in clinical field and its applications.
Despite certain presentations and methodological issues, their contribution to the field is still not well supported by the proposal, the authors may put their contribution in the context of literatures and illustrate the advantages more specifically.
The authors are encouraged to proofread the paper and improve the readability.
Based on the above reasons, I acknowledge the merits of this manuscript but would recommend a throughout revision before publication.
Reviewer 2 Report
Overall, I don't see anything new presented in this work that is appropriate for this journal. The original work in designing GOALS was novel. But this paper seems to be just about integrating a pressure measuring treadmill, which biomechanics researchers have been integrating with their camera systems for the past 20 years. I could see benefits (novelty) of the paper if the authors had focused on using the pressure data for kinetic analysis at each spinal level, but then the paper would still not be appropriate for this journal, but would be more appropriate for a biomechanics journal. Pressure treadmills and mats have been used for quite some time now to detect gait cycles and do posture/balance analysis, which seem to be the only use in this paper.
The only research this article presents is a pre-post scoliosis intervention. This case study is not novel or appropriate for this journal either.
Specific comments:
- line 35: change "to" to "us"
- line 93: remove the comma after "combine"
- line 203: change "relatively" to "relative"
- line 296: move "momentarily" to before "suspend"
- line 330-332: remove "it is visible" and move "is visible" to before "except"
- line 332: change "manifest" to "present"
- line 353: change "also dynamic flexions" to "a dynamic flexion"
- line 367: delete the sentence starting with "Again,...) as it is not needed or link it to data (in a table or graph)
- line 403: change "joints" to "joint"
- the authors over-use, and sometimes inappropriately use, conjunctive adverbs. They should work with a native english speaker to edit the paper, as it is difficult to understand what they are attempting to say sometimes.
- line 515-519: The accuracy and precision of the cameras to identify markers is not in question in this paper is not about the GOALS system. I am interested to know the accuracy and precision to identify vertebral motions, but i imagine the authors described these in a previous paper. I see in the next paragraph the authors talk about vertebral positions, but what about motions? The accuracy and precision of the pressure data (if anything) should be the focus of the discussion here since this paper is about integrating the pressure data. But that would not be novel. The authors also confuse (use interchangeably) accuracy and precision in this paragraph.
- line 551-553: Being applied to lots of people does not mean GOALS is well-accepted. How many different clinicians are using this system? What are their resulting outcomes compared to standard practices? These questions can assist you in objectively claiming it is well-accepted.